

# Prediction of life stress on athletes' burnout: the dual role of perceived stress

Theresa Chyi[1], Frank Jing-Horng Lu[2], Erica T.W. Wang[3], Ya-Wen Hsu[4] and Ko-Hsin Chang[5]

[1] Department of Exercise and Health Promotion, Chinese Culture University, Taipei City, Taiwan
[2] Graduate Institute of Sport Coaching Science, Chinese Culture University, Taipei City, Taiwan
[3] Office of Physical Education and Sports Affairs, Feng Chia University, Taichung, Taiwan
[4] Department of Physical Education, Health, and Recreation, National Chiayi University, Chiayi, Taiwan
[5] Department of Physical Education, Chinese Culture University, Taipei City, Taiwan

## ABSTRACT

Although many studies adopted *Smith's (1986)* cognitive–affective model of athletic burnout in examining stress–burnout relationship, very few studies examined the mediating/moderating role of perceived stress on the stress–burnout relationship. We sampled 195 college student-athletes and assessed their life stress, perceived stress, and burnout. Correlation analyses found all study variables correlated. Two separate hierarchical regression analyses found that the "distress" component of perceived stress mediated athletes' two types of life stress–burnout relationship but "counter-stress" component of perceived stress-moderated athletes' general-life stress–burnout relationship. We concluded that interweaving relationships among athletes' life stress, perceived stress, and burnout are not straightforward. Future research should consider the nature of athletes life stress, and dual role of perceived stress in examining its' association with related psychological responses in athletic settings.

## INTRODUCTION

Playing sports in college as student-athletes is never an easy task for young adults. The college student-athletes are ordinary students; they have to cope with college-related demands such as preparing course works, passing exams, arranging campus life, taking care of personal health, and handling interpersonal relationships. Besides, they are competitive athletes; they have to engage in athletically related tasks such as participating in vigorous training/competitions, building coaches/teammates relationship, preventing/taking care of injuries, and meeting performance demands. In addition, if they are freshmen or transfer students they have to face many life adjustment issues, such as moving to new residence, adapting new school/social systems, or building new social relationships (*Etzel, 2009*; L. T. Huffman, 2014, unpublished data; *Lu et al., 2012*). For these reasons, student-athletes in the campus are very vulnerable to stress.

Despite stress is an inevitable part of life, research indicated that chronic stress is detrimental to physical and mental health. Many studies indicated that being overly

Corresponding author
Frank Jing-Horng Lu,
ljh25@ulive.pccu.edu.tw

exposed to daily life stress increasing heart disease (*Steptoe & Kivimaki, 2012*), suffering from gastrointestinal ulcers (*Ali & Harty, 2009*), elevating the possibility of cancer (*Valko et al., 2006*), causing asthma (*Wright, Rodriguez & Cohen, 1998*), and increasing hyperglycemia (*Bosarge & Kerby, 2013*). In regard to mental problems, research found high in stress is associated with depression (*Risch et al., 2009*), increasing hopelessness and suicide ideation (*Davidson et al., 2009*; *Glick et al., 2012*), lowering well-being (*DiBartolo & Shaffer, 2002*), decreasing performance (*Humphrey, Yow & Bowden, 2000*), increasing eating disorder (*Epel et al., 2001*), and developing burnout (*Francisco et al., 2016*; *Gustafsson & Skoog, 2012*; *Lu et al., 2016*).

In sports literature, athlete burnout is one of the prominent issues receiving much attention by researchers and is closely associated with stress. Athlete burnout is defined as "...an athlete's reaction to chronic stress..." which is characterized as *feeling physically and psychologically exhausted from the demands of training and competing, perceive a reduced sense of accomplishment, and experience sport devaluation in which they engage* (*Raedeke & Smith, 2001*, p. 283). In explaining the process of burnout, *Smith (1986)* proposed a four-stage theory of athletic burnout which contended that burnout is a reaction to chronic stress comprising situational, cognitive, physiological, and behavioral components. The first stage starts from perceived situational demands including performance demands, conflicts between training and personal schedule, overload training, expectations and pressures from others. The second stage termed cognitive appraisal by which athletes' interpret these demands; particularly individuals' cognitive appraisal of the balance between challenges and resources, and potential consequences. When athletes perceived demands surpass personal resources and consequences will be severe, the process goes to the third stage of physiological and psychological responses such as anxiety, tension, insomnia, and illness. The fourth stage is athlete burnout, which is characterized by rigid and inappropriate behavior, decreased performance and withdrawal from activity.

Many studies had adopted *Smith (1986)* framework in examining the stress–burnout relationship. Generally, past research supports the association between stress and athlete burnout. For example, in an effort to examine the antecedent and consequence of athlete burnout, *Francisco et al. (2016)* investigated 453 Spanish athletes and measured their perceived stress, depression, and burnout. Results found perceived stress accounted for 43% of the variance of burnout, and perceived stress and burnout jointly accounted for 50% of the variance of depression. Similarly, by using a qualitative approach, *Gustafsson et al. (2008)* interviewed 11 Swedish athletes about the sources of burnout. They found multiple demands such as "too much sport," "lack of recovery" and "high expectations" were considered as primary causes of burnout. Further, in a search of sources and characteristics of athlete burnout, *Cresswell & Eklund (2006)* interviewed 15 New Zealand professional rugby players found many negative experiences such as injury, perceptions of the team environment and training loadings associated with burnout. Recently, *Gustafsson & Skoog (2012)* examined whether athletes high in optimism negatively predicted burnout, they measured participants' optimism, perceived stress, and burnout. Results found a mediating effect of perceived stress on the optimism–burnout

relationship. Moreover, *Gustafsson et al. (2013)* investigated the relationships among hope, perceived stress, and found perceived stress mediated the hope–burnout relationship.

Despite these efforts in examining the stress–burnout relationship, very few studies include both life stressors and global perceived stress in *Smith's (1986)* cognitive–affective model of athletic burnout. According to *Smith (1986)*, athletes encounter many situational demands from life, training, and competition. These demands are life stressors including engaging in heavy training, meeting performance demands, preventing/taking care of the injury, and building/maintaining an interpersonal relationship. Another type of stress is the global perception of the stress. That is, how athletes perceived these stressors are stressful. *Smith (1986)* contends that when athletes encounter life stressors they would engage in the cognitive appraisal. To measure this type of global perception of stress, most researchers (*Gustafsson & Skoog, 2012*; *Gustafsson et al., 2013*; *Raedeke & Smith, 2004*) used *Cohen, Kamarack & Mermelstein (1983)* perceived stress scale (PSS) to assess athletes' perceived stress. Generally, they used a composite score of PSS by reversing all the positive items of PSS and added them to all negative items. Although this approach is simple and straightforward, it omitted lots of information of how perceived stress influence athletes' psychological responses. According to *Cohen, Kamarack & Mermelstein (1983)*, perceived stress includes both positive and negative components (i.e., counter stress and perceived distress). The "counter stress" represents one's confidence in how he/she can cope any challenge/disturbance that one encounters in life. In contrast, "perceived distress" refers to one's perception of how life's situations are uncontrollable, unexpected, overloading and make one feels distressing and annoying.

Under such consideration, when examining stress–burnout relationship if researchers include two types of stress (i.e., life stress and perceived stress) and differentiate the positive/negative components of perceived stress on the life stress–burnout relationship, they would be able to explore more knowledge about the interweaving relationships among athletes' life stress, perceived stress, and burnout. Especially, recent research on the psychometric properties of PSS (*Barbosa-Leiker et al., 2013*; *Chiu et al., 2016*) confirmed that PSS comprises two factors as *Cohen, Kamarack & Mermelstein (1983)* suggested. As to life stress, *Lu et al. (2012)* developed a sport-specific life stress scale which categorized athletes' life stress into two categories—general-life stress and sport-life stress. Further, empirical studies showed individuals' counter stress cognition such as self-efficacy moderated family caregivers' behavioral problems and burnout relationship (*Romero-Morno et al., 2011*), or moderated workers' stress appraisal–quality of life relationship (*Prati, Pietrantoni & Cicognani, 2010*), and moderated the relationship between cancer patients' treatment information and well-being (*Namkoong et al., 2010*).

In contrast, the empirical studies have found perceived distress mediated the relationship between individual personality (e.g., hope and optimism) and burnout relationship as earlier mentioned works by *Gustafsson & Skoog (2012)* and *Gustafsson et al. (2013)*. In non-sport settings, it was found when nursing students perceived more stress from teaching and learning and other organizational demands they perceived high distress, which in turn led to burnout. Thus, perceived stress mediated the demands–burnout relationship. Moreover, in a large-scale investigation

(Behavioral Risk Factor Surveillance System) with a total of 85,130 participants which examined the relationship among perceived discrimination, psychological distress, and smoking status. Results found regardless of race/ethnicity, psychological distress mediated the discrimination-smoking association (*Purnell et al., 2012*).

Unfortunately, to the best of our knowledge, no researchers attempted to adopt such considerations in examining the interweaving relationships among two types of life stress: perceived stress and burnout. By using *Smith (1986)* model as a guiding framework, it is considered that the negative component of perceived stress (i.e., perceived distress) would play a mediating role in the two types of life stress–burnout relationship because perceived distress stands in the middle between environmental challenges and athlete burnout. As *Barron & Kenny (1986*, p. 1176*)* suggested, a mediator is a third variable that explains how external physical events take on internal psychological significance. In contrast, it is considered that a positive component of perceived stress (i.e., counter stress) would play a moderating role in the two types of life stress–burnout relationship because counter stress represents athletes' cognitive appraisals that they can handle stress. According to *Barron & Kenny's (1986*, p. 1174*)*, the moderator is the third variable that affects the direction/strength of independent/predictor variable and dependent/criterion variable.

## PURPOSES AND HYPOTHESES

Building on the above literature, the purpose of this study was to examine the relationships among athletes' two types of life stress, the two components of perceived stress, and burnout; and examine the mediating/moderating role of perceived stress on the life stress–burnout relationship. We hypothesized that athletes' two types of life stress, perceived stress (i.e., perceived distress and counter stress), and burnout would be significantly correlated. Further, "perceived distress" would mediate the two types of life stress–burnout relationship but "counter stress" would moderate two types of life stress–burnout relationship.

## METHODS

### Participants

Participants in this study were 195 athletes (male = 138, female = 57) with average ages of 19.89 (SD = 1.34) years recruited from two sports colleges and four universities in Taiwan. They had been participating in intercollegiate sports such as basketball, volleyball, and baseball with 7.56 ± 2.83 years of sports experiences. They averagely trained 3.54 h per day (SD = 1.32) and 5.02 days per week.

### Measurements and procedure

Prior to data collection, we gained approval from the Institutional Review Board of a hospital ethical committee. Then, we contacted athletes with the permissions of coaches and administrators. In an appointed date we arrived at their training venues by the introduction of their coaches. Then, we briefly introduced ourselves and informed participants the purpose of the research, confidentiality, and anonymity of their participation. Those who interested in this study then signed informed consent and

completed a survey package including the demographic questionnaire, the life stress, perceived stress and burnout measures. It took approximately 20 min to complete. The questionnaire administration was either before or after each team's training session. The measures as follow:

Demographic questionnaire: We used a demographic questionnaire to collect participants' age, gender, types of sports, daily training hours, training frequency per week, and years of athletic experiences.

### Athlete burnout questionnaire (ABQ)

Developed by Raedeke & Smith (2001), the athlete burnout questionnaire (ABQ) was used to assess participants' athletes' burnout experiences. The ABQ has three subscales: (a) reduced the sense of athletic accomplishment (RA); (b) perceived emotional and physical exhaustion (E), and (c) devaluation of sports participation (D). Participants identify their athletic burnout experiences using a six-point Likert scale that ranged from 1 (never) to 6 (always). Sample question for emotional/physical exhaustion is "I feel so tired from my training that I have trouble finding energy to do another thing;" for reduced sense of accomplishment is "I'm accomplishing many worthwhile things in sport;" for sport devaluation is "The effort I spend in sport would be better spent doing other things." The higher the number as participants identified as the higher the degree of burnout of the sport. The internal consistency of the present study was 0.85, 0.86, and 0.63. In this research, we used a composite score by adding three subscales together.

### College student-athletes' life stress scale (CSALSS)

We used 24-item college student-athletes' life stress scale (CSALSS) (Lu et al., 2012) to assess participants' perceptions of their daily life stress classified as general-life and sport-specific stress. The questionnaire asked questions such as "I am annoyed with my coach's bias against me," or "I worry about my unstable competitive performance." There are eight factors in the 24-item CSALSS: (a) sports injury, (b) performance demand, (c) coach relationships, (d) training adaptation, (e) interpersonal relationships, (f) romantic relationships, (g) family relationships, and (h) academic requirements. Participants indicated the frequency of the specific life event on a six-point Likert scale ranging from 1 (Never) to 6 (Always). Cronbach's α of these factors ranged from 0.70 to 0.87 and the reliability for all items was 0.93 in this study, indicating that the measure was reliable. According to Lu et al. (2012), college student-athletes life stress can be categorized as sport-specific life stress (i.e., sports injury, performance demand, coach relationships, training adaptation) and general-life stress (i.e., interpersonal relationships, romantic relationships, family relationships, academic requirements). We used the composite score of sport-specific life stress and general-life stress for statistical analyses.

### The perceive stress scale (PSS)

Developed by Cohen, Kamarack & Mermelstein (1983), the perceive stress scale (PSS) measure is used to assess one's perception of the degree of a given situation in life is stressful. We used a two-factor 10-item PSS for our study (Chiu et al., 2016). Chiu et al. (2016)

reported that two-factor 10-item PSS with appropriate psychometric properties. The sample question of "perceived distress" is (e.g., how often have you felt upset because of something that happened unexpectedly?), and sample question of "counter stress" is (e.g., How often have you felt confident about your ability to handle personal problems?). According to *Chiu et al. (2016)* items 1, 2, 3, 6, 9, 10 represent perceived distress; items 4, 5, 7, 8 represent counter stress. Participant rated their experiences of stress by answering items with a five-point Likert scale (0 = never, 1 = almost never, 2 = sometimes, 3 = fairly often, 4 = very often). We used a composite score of perceived distress and counter stress by adding all 6/4 items together.

## Statistical analyses

Firstly, the descriptive statistical analysis examined the properties of the collected data, which including skewness, kurtosis, means and standard deviations. Also, we used Pearson's product–moment correlation analysis to examine the correlations of all variables.

### Examination of mediation

For a mediating effect, we used a simple regression to examine whether the independent variable predicts mediator; mediating variable predicts dependent variable, and independent variable predicts the dependent variable. This analysis was conducted as a prerequisite analysis for testing mediating effects (*Barron & Kenny, 1986*, p. 1176). According to *Barron & Kenny (1986)*, to examine mediation effect, the following conditions should be met in our study: (a) two types of life stress should be able to account for variance in perceived distress; (b) perceived distress should be able to account for variance in burnout; and (c) two types of life stress should be able to account for variance in burnout. If all three conditions were met, the subsequent mediating of perceived stress on the relationship between two types of life stress and burnout were further analyzed. To examine the main effects, two types of life stress (i.e., sport-specific and general-life stress) were entered into the regression in the first step. Perceived distress was entered in the second step. Two types of life stress and perceived distress were entered in the third step. The final test of mediation was to examine whether two types of life stress would still predict burnout when perceived distress was controlled. Finally, a Sobel test ($Z \geqq 1.96$, $p < 0.05$) was used to examine the significance of mediating effect.

### Examination of moderation

To examine moderating effect, we used hierarchical regression analysis to examine the unique and joint contributions of two types of life stress and counter stress in predicting athlete burnout. The control variable (i.e., sex) was entered into the regression first. In the second and third steps, we examined the main effects of two types of life stress and counter stress on burnout. Finally, the full model with interaction effects of two types of life stress and counter stress were tested. Based on procedures recommended by *Cohen et al. (2002)* we graphed all significant interactions to show the relationship between counter stress and burnout using data of one standard deviation above and below the mean for two types of life stress.

**Table 1 Correlation matrix and descriptive statistics for study variables.**

|  | 1 | 2 | 3 | 4 | 5 |
|---|---|---|---|---|---|
| 1. [a]ABQ | 0.90[b] | 0.33** | 0.50** | 0.37** | −0.29** |
| [a]CSALSS |  |  |  |  |  |
| 2. General-life stress |  | 0.88[b] | 0.65** | 0.34** | −0.11 |
| 3. Sport-life stress |  |  | 0.88[b] | 0.49** | −0.23** |
| PSS |  |  |  |  |  |
| 4. Distress |  |  |  | 0.78[b] | −0.13 |
| 5. Counter stress |  |  |  |  | 0.67[b] |
| Mean | 3.03 | 2.56 | 3.08 | 2.02 | 2.03 |
| SD | 0.86 | 0.06 | 0.06 | 0.67 | 0.70 |
| Skewness | −0.28 | −0.35 | −0.30 | −0.16 | −0.55 |
| Kurtosis | 0.22 | 0.37 | −0.02 | 0.10 | 0.16 |

Notes:
[a] ABQ, athlete burnout questionnaire; CSALSS, college student-athlete life stress scale.
[b] Cronbach's α for each subscale is displayed on the diagonal.
** $p < 0.01$.

## Results

### Preliminary analysis

Descriptive statistics found skewness ranged from −0.01 to 0.36 and kurtosis ranged from −0.15 to −0.34, indicating that study variables did not exceed ±2, and all variables were in an acceptable range of normality (*Hair et al., 2006*). In addition, no outliers were found. Pearson correlation test revealed all variables were positively correlated, the coefficient between 0.23 and 0.65 as Table 1 indicated. Athlete burnout had a higher correlation with sport-specific life stress than general-life stress and perceived distress. But burnout negatively correlated with counter stress. Further, according to *Barron & Kenny's (1986)* suggestion of the prerequisite of mediation effect, a simple regression analysis showed that sport-specific stress and general-life stress positively predicted perceived distress (β = 0.49, $p < 0.00$; β = 0.34, $p < 0.00$) and burnout (β = 0.50, $p < 0.00$; β = 0.33, $p < 0.00$). Also, perceived distress positively predicted burnout (β = 0.37, $p < 0.00$) as Table 2 indicated. Therefore, it is suitable for subsequent mediation analysis.

### Mediating effects of perceived distress on life stress–burnout relationship

We used hierarchical regression analyses to examine mediation effects. As Table 3 (left) shown when perceived distress was controlled in the first step, general-life stress predicted burnout ($R^2 = 0.11$, $F(1, 194) = 23.77$). However, when perceived stress and general-life stress simultaneously entered in model 2, the prediction of general-life stress on burnout was significantly less after controlling perceived distress (β from 0.33 to 0.23). The Sobel test found that the mediating effect was significant (Sobel's $Z = 3.84$, $p < 0.05$). Similarly, as Table 3 (right) illustrated when perceived distress was controlled in the first step, sport-specific life stress predicted burnout ($R^2 = 0.25$, $F(1, 194) = 65.30$). However, when sport-specific life stress and perceived distress simultaneously entered in

**Table 2 Simple regression of life stress and perceived distress on burnout.**

| Variables | PSS-distress | | Burnout | |
|---|---|---|---|---|
| | β | $\Delta R^2$ | β | $\Delta R^2$ |
| **Regression 1[a]** | | | | |
| Sport stress | 0.49** | 0.24** | | |
| General stress | 0.34** | 0.11** | . | |
| **Regression 2[b]** | | | | |
| Sport stress | | | 0.50** | 0.25** |
| General stress | | | 0.33** | 0.11** |
| PSS-distress | | | 0.37** | 0.14** |

Notes:
[a] Dependent variable is PSS-distress.
[b] Dependent variable is Burnout.
** $p < 0.01$.

**Table 3 Mediating effects of PSS-distress on the life stress–burnout relationship.**

| | Burnout | | | | | Burnout | | | |
|---|---|---|---|---|---|---|---|---|---|
| | Step 1 | | Step 2 | | | Step 1 | | Step 2 | |
| | B | β | B | β | | B | β | B | β |
| Constant | 2.21** | | 1.70** | | Constant | 1.50** | | 1.31** | |
| Life Stress-G | 0.32** | 0.33 | 0.22** | 0.23 | Life Stress-S | 0.50** | 0.50 | 0.42** | 0.42 |
| PSS-distress | | | 0.38** | 0.29 | PSS-distress | | | 0.22* | 0.17 |
| $R^2$ | 0.11 | | 0.19 | | $R^2$ | 0.25 | | 0.27 | |
| Adjusted $R^2$ | 0.11 | | 0.18 | | Adjusted $R^2$ | 0.25 | | 0.27 | |
| Changed in $R^2$ | | | 0.08 | | Changed in $R^2$ | | | 0.02 | |
| Sobel's Z | 3.84* | | | | Sobel's Z | 4.60* | | | |

Notes:
Life Stress-G, general-life stress; Life Stress-S, sport-specific life stress.
* $p < 0.05$.
** $p < 0.01$.

model 2 the prediction of sport-specific life stress on burnout was significantly less after controlling perceived distress (β from 0.50 to 0.42). The Sobel test found the mediating effect was significant (Sobel's $Z = 4.60$, $p < 0.05$).

### *Moderating effects of counter stress on life stress–burnout relationship*

The predictive value of life stress and perceived stress on burnout is illustrated in Table 4. The main effects of two types of life stress and counter stress were significant. However, only the interaction between general-life stress and counter stress significantly predicted burnout; the interaction uniquely accounted for 3% of the variance. Based on procedures recommended by Cohen and colleagues, the graph of the interaction illustrated that for participants with high counter stress (one standard deviation above the mean) their burnout was lower than those with low counter stress (one standard deviation below the mean) (Fig. 1). The full model accounted for 19.0% of burnout. However, the interaction between sport-specific life stress and counter stress did not significantly predict burnout.

**Table 4 Summary results of the moderating effects.**

| | LS-G × PSS-counter | | | LS-S × PSS-counter | | |
|---|---|---|---|---|---|---|
| | Predictor | $R^2$ | B | Predictor | $R^2$ | β |
| Step 1 | | 0.12** | | | 0.26** | |
| | Sex | | 0.10 | Sex | | 0.09 |
| | LS-G | | 0.35** | LS-S | | 0.51** |
| Step 2 | | 0.17** | | | 0.29** | |
| | Sex | | 0.05 | Sex | | 0.05 |
| | LS-G | | 0.31** | LS-S | | 0.47** |
| | PSS-C | | −0.24** | PSS-C | | −0.17* |
| Step 3 | | 0.21** | | | 0.29** | |
| | Sex | | 0.04 | Sex | | 0.05 |
| | LS-G | | −0.21 | LS-S | | 0.28 |
| | PSS-C | | −0.71 | PSS-C | | −0.28* |
| | LS-G × PSS-C | | 0.68** | LS-S × PSS-C | | 0.27 |
| Total $R^2$ | | | | | | |

**Notes:**
LS-G, general-life stress; LS-S, sport-specific life stress; PSS-C, PSS counter stress.
* $p < 0.05$.
** $p < 0.01$.

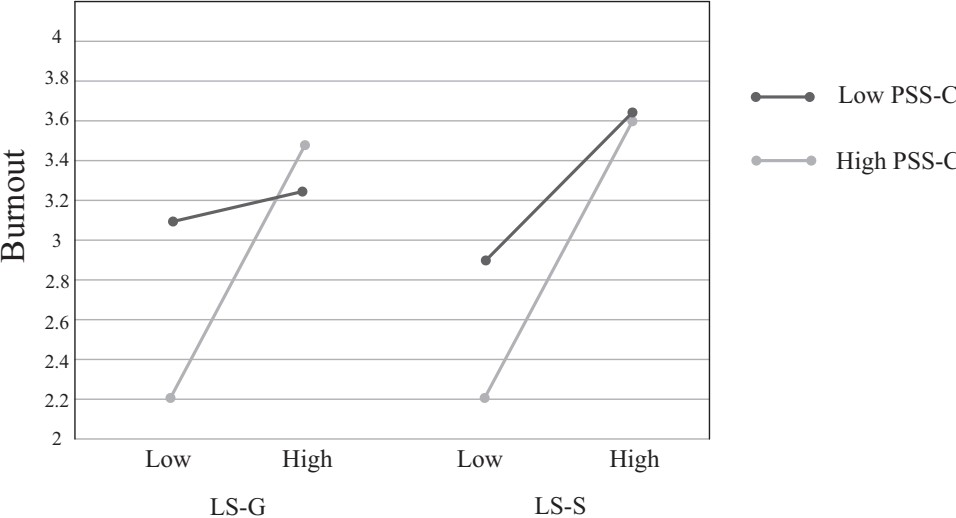

**Figure 1 The relationship between two types of life stress and burnout moderated by counter stress.** For participants with high counter stress (light line), burnout was significantly lower than participants with low counter stress in general-life stress condition but not in sport-specific life stress condition; LS-G, general-life stress; LS-S, sport-specific life stress.

# DISCUSSION

## Theoretical contributions/implications

In considering that stress plays an important role in athlete burnout, we adopted *Smith (1986)* cognitive–affective model of athletic burnout and re-examined the

triangular relationships among athletes' life stress, perceived stress, and burnout. Results found all study variables were correlated, and perceived stress plays a dual role in mediating and moderating the life stress–burnout relationship. The preliminary results provide several implications as follow:

First, to differentiate the nature and role of stress in the stress–burnout relationship adds our knowledge that both life stress and perceived stress play a different role in athlete burnout. The global perception of stress can be a mediator and moderator in the life stress–burnout relationship. The mediating effect of perceived distress supported past research that this type of stress mediated the relationship between hope and burnout (*Gustafsson et al., 2013*), optimism and burnout (*Gustafsson & Skoog, 2012*), and coping source of burnout (*Raedeke & Smith, 2004*). The moderating effect of counter stress also supported that individuals' beliefs that they can handle stress moderated the job demand–burnout relationship (*Salanova, Peiro & Schaufeli, 2002*), athletes life stress–burnout relationship (*Lu et al., 2016*), or soldiers' stressor–strain relationship (*Jex et al., 2001*).

Further, the finding of mediating and moderating effects of perceived stress added our knowledge of how life stress and perceived stress influence athletes' burnout responses. For mediating effect of perceived distress on the life stress–burnout relationship, it was found both "general-life" stress and "sport-specific" life stress and burnout relationships were all mediated by perceived distress. However, in the moderating condition, the only general-life stress–burnout relationship has been moderated by counter stress. The reason for such distinction was unknown. However, we inferred that it might be because our participants were sampled from Division-I athletes. Research indicated that experienced athletes have been taught to use all types of coping, such as increasing efforts, suppressing competing activity, active coping, and seeking social support, for stressful sports situations (*Crocker & Graham, 1995*; *Pensgaard & Ursin, 1998*). Therefore, the relationship between sport-specific life stress and burnout can't be moderated by counter stress. However, such an explanation was just referred from literature. Future studies should empirically examine the relationships among athletes' sport-specific life stress, counter stress, and burnout.

Further, the prediction of life stress on athlete burnout supported that many life stressors associated with burnout (*Cresswell & Eklund, 2006*; *Francisco et al., 2016*; *Gustafsson et al., 2008*; *Lu et al., 2016*). Specifically, this study found sport-specific life stress had higher correlation than general-life stress with burnout. This implied that athletes perceived high in sport-specific life stress are disadvantageous to burnout. Our results prompt coaches, sports administrators, parents, and sports professionals pay attention to athletes' sport-specific life stress because it may influence athletes' well-being in sports.

The athlete burnout is the core issue of the present study. Our results found athlete burnout not only associated with life stress but also mediated/moderated by different type of perceived stress. The dual role of perceived stress on the life stress–burnout relationship informs us that athletes' cognitive appraisal will influence how athletes react to life stressors. According to *Lazarus & Folkman (1984)* and *Lazarus (1993)*, any event that has

been evaluated as stressful is because individuals perceived they are lack of coping resources and abilities. If they perceived they are confident to handle these challenges they would not feel stressful. Also, if they perceived that they have many resources to cope with stressors they do not feel stressors are threatening. We suggest that future study may adopt this dual role perspective of perceived stress in other areas such as how perceived stress influences athletic injury (*Andersen & Williams, 1988*), doping (*Hodge et al., 2013*), substance abuse (*Grossbard et al., 2009*), and eating disorder (*Dockendorff et al., 2012*).

### Limitations and future suggestions

Several methodological and interpretive issues need to be discussed. First, although we found there is a moderating/mediating effect of perceived stress on the life stress–burnout relationship. Because the present study is a cross-sectional study it does not warrant cause-and-effect. Future study may adopt a longitudinal design to observe how athletes' life stress influences stress appraisals, and how perceived stress subsequently influences life stress–burnout relationship. Second, our sample was all recruited from Taiwan and Division-I college student-athletes whether our results can be generalized to other culture, or athletes with different levels such as professional athletes or younger athletes, need to be further examined. Third, in this study, we only measured participants' life stress. Whether another type of life stressors (e.g., organizational stressor) predicted athletes' burnout should be further examined (*Arnold, Fletcher & Daniels, 2013*). Moreover, although we adopted *Smith (1986)* cognitive–affective model of athletic burnout in examining stress–burnout relationship we did not include psychological/ physiological responses which are conceptualized in the third stage. Future study may include these variables in the research model and examine how they are correlated.

## CONCLUSION

This study has shown interweaving relationships of life stresses, perceived stress, and burnout in college student-athletes. The negative component of perceived stress played as a mediator both in general-life and sport-life stress–burnout relationship. On the other hand, the positive component of perceived stress played as a moderator between general-life stress–burnout but not in sport-specific life stress condition. In order to promote total health and wellness of student-athlete, we suggest that sports administrators, coaches, and parents should work together to reduce athletes' stress via effective life-management programs. By doing so, they can build a healthy athletic community and promote athletes' psychological well-being.

### Funding

This work was supported by the Ministry of Science and Technology (Taiwan) MOST 104-2410-H-179-009. The funders had no role in study design, data collection and analysis, decision to publish, or preparation of the manuscript.

## Grant Disclosures

The following grant information was disclosed by the authors:
Ministry of Science and Technology (Taiwan) MOST: 104-2410-H-179-009.

## Competing Interests

The authors declare that they have no competing interests.

## Author Contributions

- Theresa Chyi wrote the paper, reviewed drafts of the paper.
- Frank Jing-Horng Lu conceived and designed the experiments, analyzed the data, wrote the paper, reviewed drafts of the paper, supervise all the process of the research.
- Erica T.W. Wang performed the experiments, analyzed the data, contributed reagents/materials/analysis tools, prepared figures and/or tables.
- Ya-Wen Hsu performed the experiments, analyzed the data, prepared figures and/or tables.
- Ko-Hsin Chang conceived and designed the experiments, performed the experiments, analyzed the data, contributed reagents/materials/analysis tools, wrote the paper.

## Supplemental Information

Supplemental information for this article can be found online at http://dx.doi.org/10.7717/peerj.4213#supplemental-information.

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
