# Peer review of "Prediction of life stress on athletes’ burnout: the dual role of perceived stress"

_PeerJ, doi:10.7717/peerj.4213_

## Round 0.1 · original submission · Major Revisions

I have now received three reviewers' comments. Although all expressed their interest in your study, several aspects of this manuscript should be revised to improve its clarity. Their observations are presented with clarity so I'll not risk confusing matters by belaboring or reiterating their comments. While I might quibble with the occasional point, I note that I regard the reviewers' opinions as substantive and well-informed. I believe that all of the highlighted reservations require contemplation and appropriate attention in revising the document if it is to contribute appropriately to Peer J and the extant literature. Please revise or refute according to the two reviewers' comments and provide a point by point reply in addition to the revised manuscript. In addition, both reviewers also pointed out the language issue that dramatically impaired the quality of your manuscript. Therefore, I'd suggest you to have your revised manuscript gone through a thorough language editing by a professional native speaking editor before your resubmission.

Tsung-Min Hung, Ph.D.
PeerJ editor
Distinguished professor
Department of Physical Education
National Taiwan Normal University

·

Basic reporting

1. The English language is communicable. However, there are many grammar mistakes and typos to be corrected.
2. The references are generally up-to-dated. There are several APA related mistakes to be corrected in your reference list.
3. The articles structure, figs, and tables are in a good shape. Raw data are provided.
4. The corresponding results are presented to address the proposed hypotheses. The proposed hypotheses should be back up with relevant literature.

Experimental design

1. This is not an intervention/experiment. A cross-sectional survey design was used. The cross-sectional data, as acknowledged by the authors, limit the inferences of their research findings.
2. I am not sure why it is necessary to examine the relationships among life stress, perceived stress and burnout. In particular, life stress and perceived stress are highly correlated or closely related to each other. Without a doubt, perceived stress will mediate the relationship between life stress and burnout. It may be also possible (to test the alternative model) that life stress is a mediator between perceived stress and burnout.
3. Some background information about athletes' levels (classifications) in Taiwan may be provided under the "Participants" section.
4. The overall internal reliability of ABQ can be included.
5. Baron and Kenny (1986)'s approach to examine mediation effects has been criticized (Zhao, X., Lynch Jr, J. G., & Chen, Q. (2010). Reconsidering Baron and Kenny: Myths and truths about mediation analysis. Journal of Consumer Research, 37(2), 197-206.). Please use either bootstrapping or Sobel test to check the mediation effects.
6. How the interaction terms were created in your moderation analyses?

Validity of the findings

1. If you take my comments about your mediation analyses, please update the findings of mediation analysis accordingly.
2. I am not sure why other demographic items were not controlled (entered as co-variates) in your regression analyses? Have you checked whether they were correlated to your major study variables?
3. Please state if your findings support your hypotheses or not.

Additional comments

1. Line 77-78: Is it necessary to include these references here?
2. Line 115-118: Include the page number if you used a direct quo here?
3. Line 167-182: Your literature review is weak in developing your hypotheses. Consider citing relevant research to back up your hypotheses.
4. Line 194: You don't need the "plus and minus" symbol here?
5. Line 207: Do you mean "measures"? Measurement refers to the data collection process.
6. Line 285: You need a reference here.
7. Line 292: Replace "Beta" with its symbol.
8. Line 342-383: I would suggest you to cut down this section as they are not directly related to your findings.
9. Line 385-390: It is good to go beyond your findings to make a conclusion. However, it does not read like a conclusion here.
10. Table 1: Demographic items can be included. Put your notes under the table.

·

Basic reporting

See my suggestions on the Discussion Section about some detailed comments. Other sections are fine.

Experimental design

Research questions and hypotheses are well presented. Data analysis conducted in a good way.

Validity of the findings

Data is robust and statistically sound.

Additional comments

A well-conducted study and the steps of data analysis are conducted in a nice way. I have some suggestions for authors to consider that may improve their manuscript.

1. Lines 76-80: For the abstract, I would like to suggest the authors make it simple and only mention the aim of the study like “Building on the Smith (1986) cognitive-affective model of athletic burnout, the current study examine the relationship….”

2. Line 85: For the abstract, I would like to suggest not say BUT “but “counter-stress” component of perceived stress moderated”, use the word “while” or “and”.

3. Lines 188-189: Please remove the double quotation marks from “perceived distress” and “counter stress” or use the acronyms like PD and CS to represent the key words if not that many.

4. Line 325, I suggest authors remove the subheading of “Theoretical contributions/implications” from the manuscript. It is obvious in the discussion section.

5. Line 325, in the first paragraph of discussion, the authors mention: “The preliminary results provide several implications as follow…” Then paragraph starts like first second. My suggestion is that (a) in the first paragraph briefly revisit the main findings of the current study and remove the last sentence; (b) Do not say first, further, further. Just start with the topic sentence will be fine.

6. Lines 326-383: Switch the order of applied implications with limitations and future directions.

7. Lines 342-383: In the applied implications, I noticed that the findings of the current study is not been talked that much. However, I suggest the authors could think a little bit more to link each of the applied implication suggestions with the current findings. Just need an angle to cut in.

8. Lines 386-387: I suggest the conclusion can be rephrased, currently it reads like part of the suggestion rather than a conclusion of the whole study. It should be a highly summarized findings of the current study.

Reviewer 3 ·

Basic reporting

Simple English, easy to understand.
In the questionnaire uploaded, suggesting to add English version together with Mandarin language so that other researcher could read the questionnaires.

Experimental design

Sufficient statistical analyses according to the research questions stated. Objectives achieved.

Validity of the findings

The data is valid and statistically sound.

Additional comments

I would like to congratulate the authors for conducting such an exciting study, looking on the other aspect of Smith's model on the perceived stress among college student athletes.

I have attached some edits, and repeat them here:

Abstract:
Reference: Lundquist, & Wagnasson, 2013.
Two separate hierarchical regression analyses found that the….
Line 90 – Delete “On the one hand”, suggesting “The college student-athletes are
ordinary students;…..
Line 93 – Delete “On the other hand”, suggesting use “Besides”
Line 98 – arrange the reference alphabetically.
Line 127 – suggest to use “the forth stage”…
Line 129 – suggest to use “Many studies had adopted Smith (1986) …
Line 160 – suggest it omitted some information…
Line 171 – no researcher attempted to adopts …
Line 184 – building on the above literature,
Line 185 – the two components of
Discussion, paragraph 2. – Further, we found several findings in the mediating
and moderating …. For example, on the mediating effect of…
Please add more information on the finding in the Discussion section.

Annotated reviews are not available for download in order to protect the identity of reviewers who chose to remain anonymous.

---

## Round 0.2 · accepted · Accept

I have now received two reviewers’ comment and both reviewers were generally satisfied with your reply and revisions from previous comments. There is one specific omission of citation info, as pointed out by a reviewer but you can fix this while in Production.

Tsung-Min Hung, Ph.D.
PeerJ editor
Distinguished professor
Department of Physical Education
National Taiwan Normal University

·

Basic reporting

Line 171: incomplete citation (Barbosa-Leiker et al., 201?)

Experimental design

NIL

Validity of the findings

NIL

Additional comments

NIL

·

Basic reporting

No comment.

Experimental design

No comment.

Validity of the findings

No comment.

Additional comments

Thanks for your revision. I have no further comments.